# *In vitro* antiproliferative and apoptotic effects of thiosemicarbazones based on (-)-camphene and R-(+)-limonene in human melanoma cells

Paula Roberta Otaviano Soares[1]☯, Débora Cristina Souza Passos[1]☯, Francielly Moreira da Silva[2], Ana Paula B. da Silva-Giardini[3], Narcimário Pereira Coelho[4], Cecília Maria Alves de Oliveira[2], Lucília Kato[2], Cleuza Conceição da Silva[3], Lidia Guillo[1]*

1 Department of Biochemistry and Molecular Biology, Institute of Biological Sciences, Federal University of Goiás, Goiânia, Goiás, Brazil, 2 Laboratory of Natural Products and Organic Synthesis, Institute of Chemistry, Federal University of Goiás, Goiânia, Brazil, 3 Department of Chemistry, State University of Maringá, Maringá, Paraná, Brazil, 4 Department of Chemistry, Federal Institute of Mato Grosso do Sul, Nova Andradina, Mato Grosso do Sul, Brazil

☯ These authors contributed equally to this work.
* guillo@ufg.br

**Data Availability Statement:** All relevant data are within the paper and its Supporting Information files.

## Abstract

A series of 38 thiosemicarbazone derivatives based on camphene and limonene were evaluated for their antiproliferative activity. Among them, 19 were synthesized and characterized using proton and carbon-13 nuclear magnetic resonance ($^1$H and $^{13}$C NMR). For initial compound selection, human melanoma cells (SK-MEL-37) were exposed to a single concentration of a compound (100 µM) for 24, 48, and 72 hours, and cell detachment was visually observed. Cell viability was determined using the 3-(4,5-dimethylthiazol-2-yl)-2,5-diphenyl-tetrazolium bromide (MTT) method. Nineteen compounds (**4, 6, 8, 11, 13, 14, 15, 16, 17, 18, 20, 22, 25, 26, 31, 3', 4', 6',** and **9'**) yielded cell viability below 20%. Subsequently, $IC_{50}$ values for these compounds were determined, ranging from 11.56 to 55.38 µM, after 72 hours of treatment. Compound **17** (o-*hydroxy*benzaldehyde (-)-camphene-based thiosemicarbazone) demonstrated the lowest $IC_{50}$ value, followed by compound **4** (benzaldehyde (-) camphene-based thiosemicarbazone) at 12.84 µM. Regarding compound **4**, we observed the induction of a characteristic ladder pattern of DNA fragmentation through gel electrophoresis. Furthermore, fluorescence, flow cytometry and scanning microscopy assays revealed morphological changes consistent with apoptosis induction. Additionally, the measurement of caspase 6 and 8 activity in cellular extracts after treatment for 2, 4, 6, and 24 hours suggested the potential involvement of the extrinsic apoptosis pathway in the mechanism of action of compound **4**. Further investigations, including molecular docking studies, are required to fully explore the potential of compound **4** and the other selected compounds, highlighting their promising role in future melanoma therapy research.

**Funding:** This study was made possible through the research fellowships awarded to PROS and DCSP by the National Council for Scientific and Technological Development (CNPq) of Brazil. These fellowships supported PROS during her master's studies and DCSP during her doctoral studies, covering personal expenses during the research period, and they did not include direct salaries. CNPq played no additional role in the study design, data collection and analysis, decision to publish, or manuscript preparation. No additional external finding was received for this study.

**Competing interests:** The authors have declared that no competing interests exist.

## Introduction

Cancer encompasses a diverse group of diseases characterized by uncontrolled cell growth, necessitating the exploration of antiproliferative strategies. Globally, cancer accounted for approximately 19.3 million new cases and nearly 10.0 million cancer-related deaths in 2020 [1]. Skin cancer, in particular, merits significant attention, as it accounts for approximately 7.9% of total cancer-related deaths. In 2020, 1,500,000 new cases were diagnosed, resulting in approximately 120,000 deaths [2].

Among skin cancers, cutaneous melanoma is an exceptionally aggressive tumor, contributing to nearly 47% of new global skin cancer deaths. The aggressiveness of melanoma arises from its capacity to spread to various organs, including the lungs, liver, brain, bones, and lymph nodes. This is coupled with its remarkable ability to evade the immune system [3]. Melanoma cells share numerous cell surface molecules with vascular cells, which enhances the tumor's angiogenic potential. Additionally, these cells exhibit a significantly greater degree of stem cell-like properties than other solid tumors. These distinctive features collectively characterize melanoma as an exceptionally challenging neoplasm.

Fortunately, therapeutic options have expanded in recent years due to a better understanding of the molecular mechanisms that lead to the malignant transformation of melanocytes [4, 5]. Consequently, new avenues of treatment, including immunotherapies and oncolytic vaccines, have emerged alongside the currently approved therapies involving monoclonal antibodies and synthetic kinase inhibitors [6–9]. Nevertheless, the development of toxicity and resistance continues to pose a significant challenge [10], emphasizing the need to explore new chemotherapeutic agents, whether in combination with immunotherapy or in conjunction with the current inhibitors targeting signaling pathways.

Thiosemicarbazones are synthetic compounds developed for the treatment of various diseases owing to their versatile activities, including antiviral, antibacterial, antiparasitic, and anticancer activities [11, 12]. Their antineoplastic activities are well documented in the literature across various tumor cell lines and some animal models [13, 14]. For instance, 3-aminopyridine-2-carboxaldehyde thiosemicarbazone (Triapine®), advanced to phase II clinical trials. However, due to adverse effects observed with that compound, extensive research has been conducted to explore other derivatives that exhibit enhanced activity and selectivity. Concurrently, investigations into their mechanisms of action have made significant progress in recent years. Beyond the well-established pathways, such as ribonucleotide reductase activity inhibition and the generation of reactive oxygen species, apoptosis has emerged as a proposed mechanism [15]. We previously chose natural products such as camphene and limonene as starting materials for synthesizing thiosemicarbazide and thiosemicarbazone derivatives with antifungal [16–18], antibacterial [19, 20] and anticancer activities [21]. In this study, we expanded the series of thiosemicarbazones containing camphene and limonene, synthesizing novel derivatives that were characterized using spectroscopic methods. As our specific focus was on identifying potential anti-melanoma agents, we utilized the human melanoma cell line SK-MEL-37 derived from cutaneous metastatic melanoma. These cells notably express numerous antigens associated with melanoma [22], including those crucial for metastasis [23], making them an invaluable cellular model for screening potential therapeutic agents against this aggressive cancer.

This work achieved its objective by demonstrating that several derivatives, including novel compounds, exhibited antiproliferative activity. Evidence of apoptosis was obtained through DNA fragmentation, flow cytometry, and scanning electron microscopy techniques. We also demonstrated the induction of caspase enzymatic activity.

## Materials and methods

### Chemistry

In this study, we investigated a total of 38 compounds, and their molecular structures are illustrated in Fig 1.

The specifications of all solvents and reagents, including the natural monoterpenes camphene **1** and limonene **1'**, used in the synthesis of the studied compounds were previously described [24–26]. The synthesis of (-)-camphene **2** and R-(+)-limonene **2'** isothiocyanates followed the methods described in the literature [27, 28], and their infrared, mass and nuclear magnetic resonance spectra were determined [18, 24–26]. The preparation and structural characterization of (-)-camphene **3** and R-(+)-limonene **3'** thiosemicarbazides were also previously detailed [18, 24–26].

The aldehyde series of camphene-based thiosemicarbazones was initiated based on the synthesis of benzaldehyde thiosemicarbazones (**4–14**, Fig 1) by Barbosa da Silva, 2010 [24]. Among these, the syntheses of benzaldehyde -(-)- camphene-based (**4**), *p-methoxy*(**6**), *o-chloro* (**7**), *p-chloro* (**9**), *p-nitro* (**12**), *p-dimethylamine* (**13**) and *p-hydroxy* (**14**) were previously published, and the compounds were evaluated for anti-*Mycobacterium tuberculosis*, *Staphylococcus aureus* and *Enterococcus* spp. activity [19, 20]. The remaining compounds (**5, 8, 10, 11**) synthesized by da Silva et al. [24], are still unpublished and are highlighted in bold in Fig 1. Next, another three compounds of the camphene-based benzaldehyde series were synthesized (**15, 16, 17;** Fig 1) [25], among which *p-fluoro* (**16**) and *o-hydroxy* (**17**) have already been evaluated for anti-*Mycobacterium tuberculosis* activity [20].

The synthesis procedures for the newly synthesized camphene-based derivatives are detailed below.

### Experimental procedures for synthesis of the new benzaldehyde (-)-camphene-based thiosemicarbazones (5, 8, 10, 11, and 15)

Equimolar amounts of the following aldehydes were added to a flask with a magnetic stirrer and containing 1.1 mmol of camphene thiosemicarbazide solubilized in chloroform: *p-tolualdehyde*, *m-chloro*benzaldehyde, *o-nitro*benzaldehyde, and *m-nitro*benzaldehyde. The mixture was stirred for 15 minutes at room temperature in the presence of a catalytic amount of concentrated hydrochloric acid. The progress of the reaction was monitored by thin layer chromatography (TLC) using a 30% hexane/ethyl acetate mixture as the eluent. Subsequently, the solvent was evaporated, and the product was recrystallized from ethanol, with a 90–95% yield. The synthesis of compound **15** was carried out using the following procedure. A total of 0.44 mmol of 3-*methoxy*-4-*hydroxy*-benzaldehyde was added to a round-bottomed flask with a magnetic stirrer. Absolute ethyl alcohol and one drop of 25% sulfuric acid were added, and the mixture was stirred for 5 min. After this period, 100 mg (0.44 mmol) of thiosemicarbazide was added to the reaction mixture. The progress of the reaction was monitored by TLC using n-hexane/ethyl acetate (7:3) as the eluent and resublimated iodine as the developer. All reactions were performed at room temperature. The solvent was rotary evaporated, and the product was purified by crystallization from absolute ethanol.

### Experimental procedures for the synthesis of the new substituted acetophenone (-)-camphene-based thiosemicarbazones (18, 19, 20, 21, 22, 23, and 24)

The ketone series of camphene-based thiosemicarbazones was prepared and characterized by Coelho, 2011 [25]. It consists of seven compounds (**18, 19, 20, 21, 22, 23, 24;** Fig 1), all of which have yet to be published. The experimental procedure is outlined below, with characterization and spectra provided in the Supporting Information (S1 File).

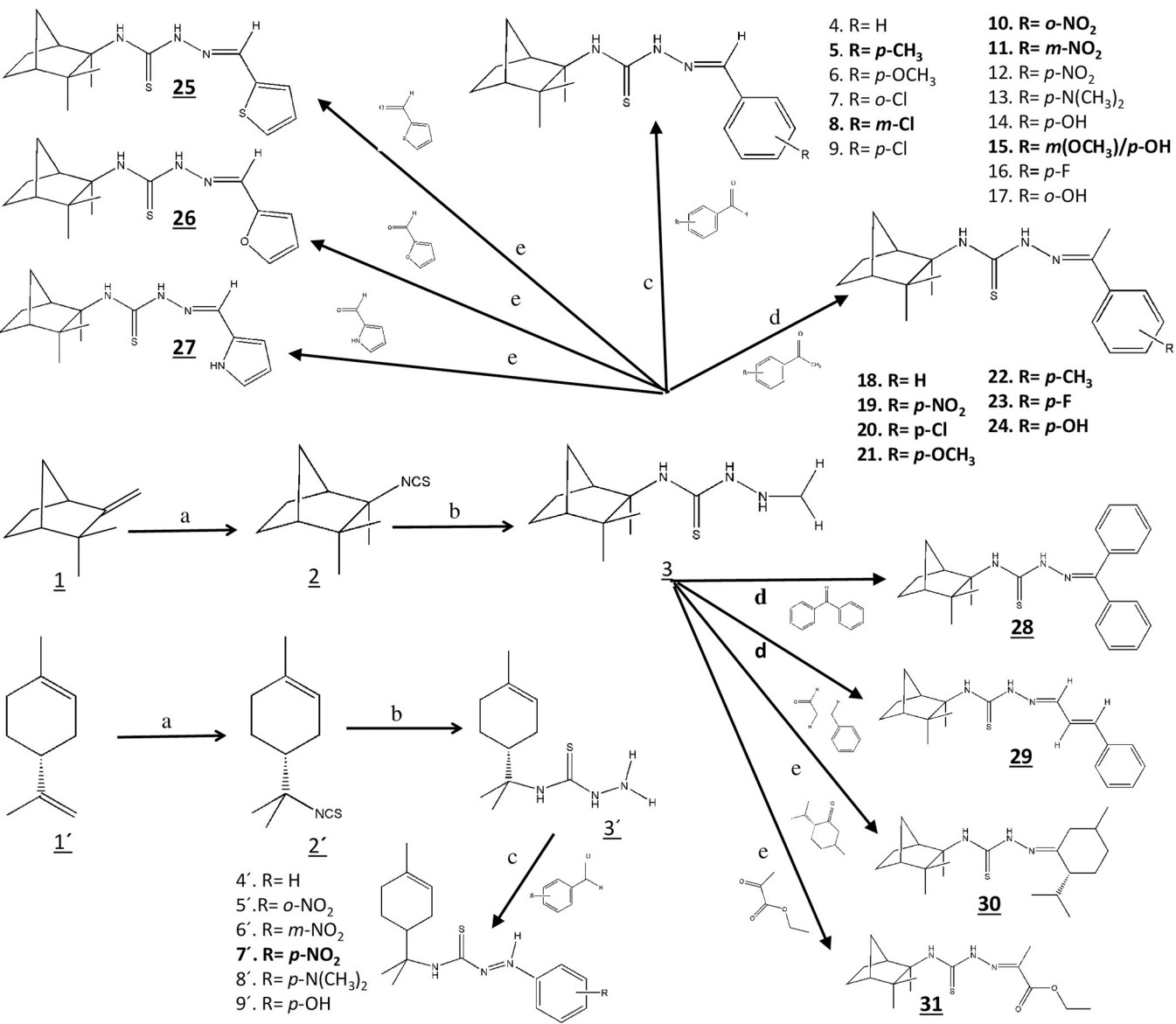

4. R= H
5. **R= *p*-CH$_3$**
6. R= *p*-OCH$_3$
7. R= *o*-Cl
8. **R= *m*-Cl**
9. R= *p*-Cl

10. **R= *o*-NO$_2$**
11. **R= *m*-NO$_2$**
12. R= *p*-NO$_2$
13. R= *p*-N(CH$_3$)$_2$
14. R= *p*-OH
15. **R= *m*(OCH$_3$)/*p*-OH**
16. R= *p*-F
17. R= *o*-OH

18. R= H
19. R= *p*-NO$_2$
20. R= p-Cl
21. R= *p*-OCH$_3$

22. **R= *p*-CH$_3$**
23. **R= *p*-F**
24. R= *p*-OH

4′. R= H
5′.R= *o*-NO$_2$
6′. R= *m*-NO$_2$
7′. **R= *p*-NO$_2$**
8′. R= *p*-N(CH$_3$)$_2$
9′. R= *p*-OH

**Fig 1. Synthetic route and reaction conditions for the target compounds.** The synthesis of the target compounds was accomplished through a series of reactions. The reactions were carried out as follows: (a) KHSO$_4$, KSCN, CHCl$_3$, at room temperature for 24 hours; (b) NH$_2$NH$_2$, HCl/NaCO$_3$/H$_2$O, in EtOH, refluxed at 90°C for 3 hours; (c) CHCl$_3$, HCl, at room temperature; (d) SiO$_2$, HSO$_4$ 5%; and (e) EtOH, HSO$_4$ 25%. All compounds whose structures are presented were tested in this study, except for compounds 1' and 2'. The bold numbers in the figure correspond to novel compounds, whose structural characterization can be found in the S1 File.

Ten milligrams of silica gel:5% H$_2$SO$_4$ and 2 mmol of one of the following carbonyl compounds were added to a test tube: acetophenone, *p-nitro*acetophenone, *p-chloro*acetophenone, *p-methoxy*acetophenone, *p-methyl*acetophenone, *p-fluoro*acetophenone, and *p-hydroxy*acetophenone. The mixture was stirred for 1 minute, and then an equimolar amount of (-)-camphene thiosemicarbazide was added. The mixture was manually shaken for a period of 5–20 minutes under microwave irradiation for 1-min intervals at minimum power. To monitor the reaction, a small amount was removed, solubilized in methanol and subjected to TLC using 7:3 *n*-hexane:ethyl acetate as the eluent and resublimated iodine as the developer. After completion of the reaction, the product was extracted with absolute ethyl alcohol and then filtered

to remove the silica gel. Thiosemicarbazones were obtained by crystallization with yields of 85–97% without the need for purification.

## Experimental procedures for the synthesis of the new camphene-based thiosemicarbazones with heterocyclic aldehyde nuclei (25, 26, and 27)

Three novel thiosemicarbazones bearing heterocyclic aldehyde moieties and derived from (-)-camphene (**25**, **26**, **27,** Fig 1) were synthesized following the method described by Coelho in 2011 [25]. The detailed experimental procedure can be found below, while characterization data and spectra are provided in the Supporting Information (S1 File). Notably, these results have not yet been published.

A total of 0.44 mmol of one of the following aldehydes, thiophene-2-carboxaldehyde, $^1$H-pyrrole carboxaldehyde, or 2-furaldehyde, was added to a round-bottomed flask with a magnetic stirrer. Absolute ethyl alcohol and one drop of 25% sulfuric acid were then added, and the mixture was stirred for 5 min. After this period, 100 mg (0.44 mmol) of thiosemicarbazide (30) was added to the reaction mixture. The progress of the reaction was monitored by TLC using n-hexane/ethyl acetate (7:3) as the eluent and resublimated iodine as the developer. All reactions were performed at room temperature. The solvent was rotary evaporated, and the products were purified by crystallization from absolute ethanol.

## Experimental procedures for the synthesis of the new camphene-based thiosemicarbazones 28, 29, 30, and 31

Thiosemicarbazones derived from (-)-camphene with the substituents benzophenone (**28**), cinnamic aldehyde (**29**), menthone (**30**), and ethyl pyruvate (**31**), all of which are novel, were synthesized based on the methodology described in [25] and have not been published to date. The experimental procedure is detailed below, with molecular characterization and spectra available in S1 File.

Ten milligrams of silica gel:5% $H_2SO_4$ and 2 mmol of one of the following carbonyl compounds were added to a test tube: benzophenone and cinnamic aldehyde. The mixture was stirred for 1 minute, and then an equimolar amount of previously ground (-)camphene thiosemicarbazide was added. The mixture was manually stirred for a period of 5–20 minutes under microwave irradiation for 1-min intervals at minimum power. To monitor the reaction, a small amount was removed, solubilized in methanol and subjected to TLC using 7:3 n-hexane: ethyl acetate as the eluent and resublimated iodine as the developer. After completion of the reaction, the product was extracted with absolute ethyl alcohol and then filtered to remove the silica gel. Thiosemicarbazones were obtained by crystallization, with yields of 85–97% without the need for purification. Compounds **30** and **31** were prepared by a procedure identical to that described for the synthesis of compounds with heterocycle aldehyde nuclei, with the addition of menthone and ethylpyruvate, respectively, to the carbonyl compound.

The series of limonene-based thiosemicarbazones derived from benzaldehyde (**4', 5', 6', 7', 8', 9'**; Fig 1) were prepared by the method described by Barbosa da Silva, 2010 [24], and have been previously published and assessed for their antitumor and antileishmanial activities [21, 29]. Details of the synthesis procedures can be found in the cited articles.

## Preparation and dilution of test compounds

Solid compounds were first solubilized in a small volume of dimethylsulfoxide (DMSO) followed by the addition of absolute ethanol (1:14, v/v). The stock concentration of all compounds was 20 mM. Serial dilutions (1:1) were then prepared in absolute ethanol, and new

stock concentrations were prepared: 10 mM, 5 mM, 25 mM, 12.5 mM, 6.25 mM and 3.125 mM. These solutions were stored at room temperature in a humidified chamber until use. A constant volume of these serial solutions was added directly to cultured cells so that the final concentration of ethanol was kept constant at 1% for all compound concentrations, and the maximum final concentration of DMSO was 0.035%. Doxorubicin hydrochloride (Sigma–Aldrich) was prepared in sterile distilled water at a 20 mM stock concentration and was used as a reference compound [30].

## Cell culture growth conditions

The human melanoma cell line SK-MEL-37 was kindly provided by Dr Lloyd Old (Memorial Sloan-Kettering Cancer Center, New York, USA) in 2000 [22]. The cells were maintained in Gibco Minimum Essential Medium (MEM) from Thermo Fisher Scientific Walthan MA, USA) supplemented with 10% (v/v) fetal bovine serum (FBS, Cultilab, Brazil) and 1% antibiotic/antimycotic (Invitrogen, USA) at 37˚C in a 5% $CO_2$ atmosphere. The cells were routinely screened for mycoplasma contamination with a MycoFluor™ Mycoplasma Detection Kit (Invitrogen, USA).

Growth characteristics were periodically checked by calculating the population doubling time (PDT) with the formula: $N = 2^z \times N_o$, where N = the cell number on the established day, $N_o$ = the cell number on day 0 (inoculation day) and z = the number of generations, where z = T/PDT. Cells were counted using a handheld automated cell counter (Scepter, Millipore) and inoculated into 25 $cm^2$ tissue culture flasks on day 0. At 24, 48, 72, 96, 100 and 120 h, the cells were harvested and counted, and the PDT was calculated. For experiments, cell densities were adjusted according to the PDT to $2.5 \times 10^4$, $1.5 \times 10^4$ and $8 \times 10^3$ cells/well for 24, 48, and 72 hours, respectively. All experiments were conducted during the exponential growth phase.

## Prescreening and cell viability studies

All tested compounds were first screened at a single dose (100 μM) in quadruplicate wells and monitored over 24, 48, and 72 hours by observing the detachment of cells with an inverted microscope. At the end of 72 h, the 3-[4,5-dimethylthiazol-2-yl]-2,5-diphenyltatrazolium bromide (MTT, Sigma–Aldrich) test (see below) was performed to quantify the percentage of detached cells in relation to that among cells treated with the solvent (control), and the values were plotted on a graph of cell viability (% of control). Cell viability was calculated by the formula: absorbance (treated cells)/absorbance (solvent-treated cells) x 100. The experiments were conducted on three independent occasions. Compounds that induced more than 80% detachment of cells (resulting in less than 20% cell viability) were chosen for further experimentation to determine their inhibitory concentration ($IC_{50}$), which is the drug concentration that reduces survival by 50%. The selected compounds were added in six different dilutions (with final concentrations of 3.125, 6.25, 12.5, 25.0, 50.0, and $100 \times 10^{-6}$ M) in quadruplicate wells. These experiments were performed on three separate occasions. After treatment, cell viability was assessed by the MTT assay, which detects mitochondrial dehydrogenase activity in viable cells [31], with slight modifications. Briefly, MTT solution was added directly to the culture medium, and the plates were incubated for 3 h at 37˚C in a 5% $CO_2$ atmosphere. The medium was aspirated, and a 300 μL aliquot of pure cold methanol was added and incubated with slight agitation for 10 minutes in the dark at RT. A 200 μL aliquot was transferred to a 96-well microplate, and absorbance readings at 570 nm were performed on a microplate reader (ELX 800, Bio-Tek Instruments, Inc.). Control cells were treated with final maximum DMSO/ethanol concentrations of 0.035% and 1%, respectively. This solvent concentration had

no cytotoxic effect on the cells. The concentration response curves were generated by plotting cell viability *versus* log concentration to obtain the inhibitory concentration ($IC_{50}$).

## DNA fragmentation analysis

SK-MEL-37 cells were seeded in 100 mm dishes and grown at 37°C until 80% confluence when they were treated with 100 μM test compounds. The treated cells were harvested at 24, 48, or 72 hours, depending on the observed cell detachment time, and lysed in buffer containing 10 mM Tris-HCl, 10 mM EDTA and 0.5% Triton X-100. The cell lysate was then treated with RNase A (100 μg/ml, final) for 30 min and proteinase K (200 μg/mL, final) for 30 min at 37°C in a water bath. Total DNA was extracted with an equal volume of saturated phenol–chloroform-isoamyl alcohol (25:24:1) (Sigma Aldrich, USA), precipitated with two volumes of cold ethanol and 0.5 M sodium chloride, and pelleted by centrifugation. The DNA pellet was resuspended in 30 μL of 10 mM Tris-HCl, 1 mM EDTA, pH 8.0, and analyzed in a 2% agarose gel containing 0.5 μg/mL ethidium bromide for visualization of the DNA fragments (DNA ladders).

## Fluorescence microscopy

**Staining with propidium iodide.**   Cells were seeded on 35 mm culture dishes and incubated in a humidified incubator with 5% $CO_2$ at 37°C until they adhered. Next, the cells were treated with compounds **4**, **8**, and **11** at a concentration of 50 μM for 6, 10 and 24 hours. Four hours after the completion of each treatment period, a propidium iodide (PI) solution was added, and the cells were monitored for growth using an inverted fluorescence microscope. Images were captured using a Nikon E6000 Eclipse fluorescence microscope. Compounds **13** and **6** were also evaluated; however, their effects were observed only after 48 and 72 h of treatment, respectively (images not shown).

*In vivo* **Caspase-3 fluorescence.**   Caspase-3 activity *in vivo* was detected using a NucView 488 Caspase-3 Kit for Live Cells from Biotium (Hayward, CA, USA) according to the manufacturer's specifications. Melanoma cells (SK-MEL-37) were seeded in 3.5 cm² petri dishes (Nunc Brand Products, USA) at a concentration suitable to reach confluency within 48 hours. Twenty-four hours after seeding, benzaldehyde (-)-camphene-based thiosemicarbazone (100 μM) was added; 8 hours later, the medium was removed and replaced with a culture medium containing NucView™ solution (1 μM). Fifteen minutes after the addition of the NucView™ solution, the cells were monitored and photographed using an inverted fluorescence microscope (Eclipse TE 2000S, NIKON) to detect caspase activity in real time.

## Caspase activity assay

Caspase 2, 3, 6, 8 and 9 activity was determined using a Caspase Colorimetric Protease Assay Kit, Apo Target™ (Invitrogen, USA). For each protease assayed, exponentially growing cells (3 x $10^5$ cells/mL) were cultured in 100-mm dishes and incubated for 24 hours at 37°C and with 5% $CO_2$ for cell adhesion. The medium was then aspirated and replaced by fresh medium containing a fixed concentration of 100 μM benzaldehyde (-)-camphene-based thiosemicarbazone (compound **4**). The cells were treated for 2, 4, 6 and 24 hours at 37°C and with 5% $CO_2$. At the end of each incubation period, cells were harvested and collected by centrifugation for 3 min at 2,000 rpm and washed twice in cold phosphate-buffered saline (PBS). The cells were transferred to a 1.5 mL microfuge tube and collected again by centrifugation for 3 minutes at 2,000 rpm. The cell pellet was resuspended in 110 μL of cold cell lysis buffer provided by the kit, the suspension was incubated on ice for 10 minutes, and the suspension was clarified by centrifugation at 12,000 rpm for 10 minutes at 4°C. Ten

microliter aliquots of the supernatant were removed for protein concentration determination. All further steps were performed according to the manufacturer's instructions, with slight modifications. To standardize the optimal incubation time for the enzymatic reaction, duplicate aliquots of 50 μL of the supernatant were combined with 50 μL of the reaction buffer containing 10 mM DTT. The mixture was then incubated in the dark at 37˚C for 2, 15, and 30 hours with 200 μM peptide substrates. The substrates were VDVAD (Val-Asp-Val-Ala-Asp- for caspase-2), DEVD (Asp-Glu-Val-Asp for caspase-3), VEID (Val-Glu-Ile-Asp for caspase-6), IETD (Ile-Glu-Thr-Asp- for caspase-8), and LEHD (Leu-Glu-His-Asp-for caspase-9), and they were labeled at their C-terminal region with the chromophore *p*-nitroaniline (*p*NA). Samples (100 μL) were transferred in duplicate to wells of a microtiter plate. Absorbance was measured at 405 nm using a microplate reader (Biotek, USA). The experiments were performed on two separate occasions. Additionally, a standard curve was plotted to calculate the concentration of the chromophore *p-nitro*aniline released by caspase activity. A 10 mM stock solution of *p-nitro*aniline in DMSO was prepared, and solutions containing 200 μM, 100 μM, 50 μM, 20 μM, 10 μM and 5 μM of the standard were made in the lysis buffer provided by the kit. The results were plotted using GraphPad Prism 4.00 for Windows (GraphPad Software, San Diego, USA). The graph of the enzymatic kinetics obtained for each caspase, as well as the standard curve, are presented in the Supporting Information (S2 File).

The protein concentration in each sample was determined using a bicinchoninic acid assay [32], with bovine serum albumin (BSA) as the standard protein. The standards were prepared by serial dilutions resulting in concentrations ranging from 200 μg/mL to 6.25 μg/mL. Each sample was diluted (1:100) in PBS (pH 7.2), and 50 μL was added to a 4% copper sulfate and BCA solution at a 1:50 ratio. The samples were transferred to triplicate wells of a microtiter plate and incubated at 60˚C for 30 minutes, followed by incubation at room temperature for 15 minutes. The absorbance of the samples was then measured at 570 nm using a microplate reader (ELX-800, Bio-Tek Instruments, Inc.).

The concentration of both *p-nitro*aniline and protein in each sample was determined using the respective standard curves generated with the GraphPad Prism program, version 4.0 for Windows (GraphPad Software, San Diego, USA, www.graphpad.com). Subsequently, the activity of caspases was calculated by dividing the concentration of *p-nitro*aniline by the concentration of protein. These values were then plotted in the graphs as μM/μg protein.

## Flow cytometric assays

**Annexin staining.** For this experiment, a TACS^TM Annexin V-FITC Apoptosis Detection Kit (R&D System) was employed according to the manufacturer's specifications. Cells were seeded in 10 cm$^2$ Petri dishes and incubated for 24 hours before being treated with either 100 μM benzaldehyde (-)-camphene-based thiosemicarbazone or 100 μM benzaldehyde R-(+)-limonene-based thiosemicarbazone. After 8 hours of treatment, the cells were detached using a cell scraper, washed with DPBS (4˚C), and collected by centrifugation at 2000 rpm for 3 minutes. The resulting pellet was washed twice with DPBS and then labeled with Annexin and PI solution (10 μL of 10x buffer, 10 μL of PI, 1 μL of Annexin V-FITC, 79 μL of Milli-Q water) and incubated for 15 minutes at room temperature. After incubation, 400 μL of 1x buffer (50 μL of 10x buffer and 450 μL of Milli-Q water) was added, and the sample was allowed to rest for 30 minutes. Acquisition was performed in a FACScan flow cytometer (Becton Dickinson, Franklin Lakes, New Jersey, USA), with 10,000 events registered. Early and late apoptotic events were analyzed using CellQuest^TM software.

## Scanning electron microscopy (SEM)

SK-MEL-37 melanoma cells were cultured on 3.5 $cm^2$ petri dishes (Nunc Brand Products, USA) coated with 0.1% gelatin. Benzaldehyde (-)-camphene-based thiosemicarbazone (100 μM) was added to the cells, and the cells were incubated for 8 hours. The cells were then fixed with Karnovsky's fixative solution at 4˚C for 72 hours, followed by washing with DPBS containing $Ca^{++}$ and $Mg^{++}$. The cells were subsequently washed twice with 0.1 M sodium caco-dylate buffer (pH 7.2) for 5 minutes each time, dehydrated with increasing concentrations of ethanol (30%-100%), and embedded in 1 mL of hexamethyldisilazane (HMDS) for 30 minutes at room temperature. The samples were coated with a thin layer of gold and imaged using a scanning electron microscope (Jeol, JSM—6610) equipped for energy dispersive spectroscopy (EDS) and with the Thermo Scientific NSS Spectral Imaging system, operating at 4 kV. As a control, the same experimental procedure was performed on SK-MEL-37 cells treated with the solvent only, consisting of DMSO/ethanol concentrations of 0.035% and 1%, respectively.

## Data calculations

The prescreening study results were analyzed using Microsoft Excel software, version 365, and are presented graphically as a percentage relative to the cells treated with the solvent, using the formula: cell viability = absorbance (treated cells)/absorbance (solvent-treated cells) x 100. For these experiments, the mean and the standard deviation (SD) for quadruplicate wells from triplicate experiments are presented. The inhibitory concentration value ($IC_{50}$) was obtained using a nonlinear regression model based on a sigmoidal dose–response curve. This calcula-tion was performed for 4 wells in triplicate samples using GraphPad Prism version 4.00 for Windows, GraphPad software (San Diego, California, USA, www.graphpad.com). Results are expressed as the arithmetic mean and standard deviation (SD) for three independent experi-ments. The caspase activity data are presented as the arithmetic mean and standard deviation (SD) from two separate experiments. Statistical analysis was performed using ANOVA fol-lowed by a *post hoc* Dunnett's test for comparisons with the control group.

## Results

### Cell culture characteristics and antiproliferative activity

First, we established the growth curve for the cell line under investigation (not shown), follow-ing the protocol outlined in the Materials and methods section. A PDT value of 30 hours was obtained, which was then used to calculate the number of cells seeded for each time point (24, 48, and 72 hours) of drug treatment. This approach ensured that the cells were maintained in the exponential growth phase during drug treatment at the different time points.

The determination of the $IC_{50}$ value is an important step in the study of cytotoxicity [24]. These investigations were preceded by experiments that assessed cell viability using a single concentration for treatment. In this study, cells were treated with a concentration of 100 μM compound and visually monitored for 24, 48 and 72 hours. After 72 hours, cell viability was determined using the MTT method. The results of this prescreening process for 38 compounds are shown in Fig 2 (left axis), and 19 compounds yielded cell viability below 20% and were selected for further $IC_{50}$ determination studies. Among these selected compounds, visual observations revealed that cell detachment occurred within different timeframes. Specifically, compound **13** (*p*-d*imethylamino*benzaldehyde (-)-camphene-based) produced cell detachment after 48 hours of treatment, whereas compound **6** (*p*-m*ethoxy*benzaldehyde (-)-camphene-based thiosemicarbazone) induced this effect after 72 hours. In contrast, the remaining com-pounds exhibited this effect within 24 hours of treatment. Fig 2 illustrates the results after 72

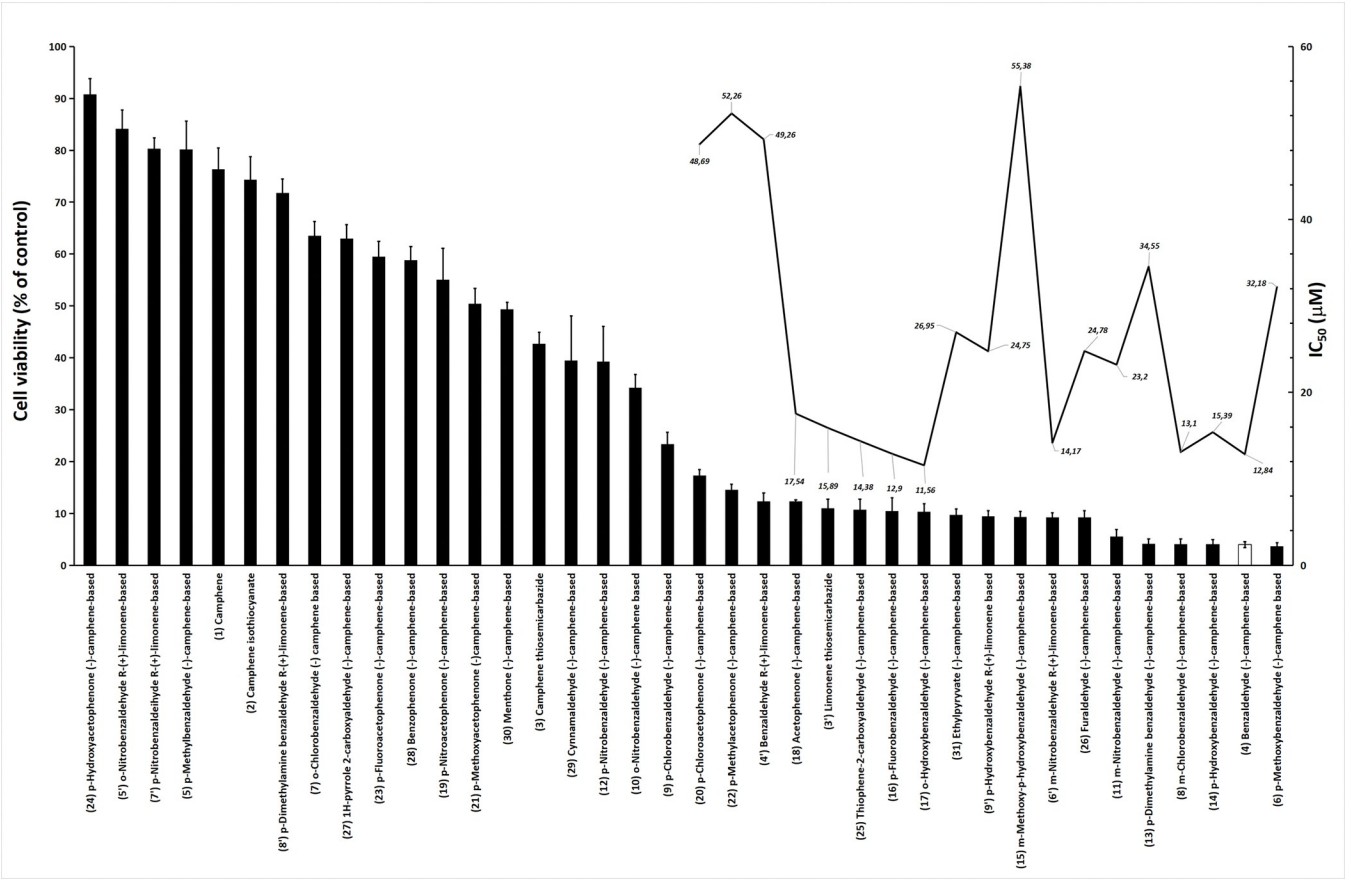

**Fig 2. Effect of camphene- and limonene-based thiosemicarbazones on the viability of human SK-MEL-37 melanoma cells.** The left y-axis indicates the percentage cell viability for the compounds. The columns in the figure represent the mean and standard deviation (SD) for quadruplicate wells from three independent experiments. Compounds inducing cell viability below 20% (on the left axis) were selected for determining IC$_{50}$ values (on the right axis). The IC$_{50}$ values shown in the figure were calculated from the mean and SD for three independent experiments. Please note that the standard deviation of IC$_{50}$ values is not displayed in the figure. The compound chosen for further experiments assessing the DNA fragmentation profile, morphological alterations, flow cytometry profile and caspase activity is represented by a blank column. The compounds named "camphene-based thiosemicarbazone" have been abbreviated as "camphene-based" on the x-axis. The number of each compound is indicated in parentheses.

hours of treatment. The IC$_{50}$ values for the 19 selected compounds are presented in Table 1 and are displayed on the right axis of Fig 2. The compounds in Table 1 are ordered by their numbering, not by their IC$_{50}$ values.

We observed that the IC$_{50}$ values were distributed between approximately 56 μM and 11 μM. Of the 19 compounds selected, 9 were novel compounds (compounds **8**, **11**, **15**, **18**, **20**, **22**, **25**, **26**, and **31**), which have not been previously reported in the literature. The remaining 10 compounds listed in Table 1 were previously evaluated for different activities, such as anti-bacterial activity [19], anti-*Mycobacterium tuberculosis* activity [20], antileishmanial activity [20], or anticancer activity against other tumor cell lines [21].

## Evaluation of apoptotic DNA ladder induction

In the following set of experiments, we aimed to investigate the mechanism of cell death induced by the selected compounds. As shown in Fig 3 and S1 Raw image, treatment with compounds **4**, **8** and **11** induced a typical DNA "laddering" pattern, which is indicative of apoptotic events, after 24 hours of continuous drug exposure. Additionally, compounds **14**, **13**,

**Table 1. Selected compounds with their respective IC$_{50}$ values obtained.**

| Compound | Number in Fig 1 | IC$_{50}$ (µM), Mean (SD) | Ref * |
|---|---|---|---|
| Benzaldehyde (-)-camphene-based | 4 | 12.84 (1.14) | [20] |
| *p-Methoxy*benzaldehyde (-)-camphene-based | 6 | 32.18 (2.01) | [20] |
| **m-Chlorobenzaldehyde (-)-camphene-based** | 8 | 13.1 (1.15) | |
| **m-Nitrobenzaldehyde (-)-camphene-based** | 11 | 23.1 (2.13) | |
| *p-Dimethylamine*benzaldehyde (-)-camphene-based | 13 | 34.55(1.22) | [20] |
| *p-Hydroxy*benzaldehyde (-)-camphene-based | 14 | 15.39 (2.77) | [19, 20] |
| **m-Methoxy-p-hydroxybenzaldehyde (-)-camphene-based** | 15 | 55.38 (3.03) | |
| *p-Fluoro*benzaldehyde (-)-camphene-based | 16 | 12.90 (1.07) | [20] |
| *o-Hydroxy*benzaldehyde (-)-camphene-based | 17 | 11.56 (1.21) | [20] |
| **Acetophenone (-)-camphene-based** | 18 | 17.54 (1.52) | |
| **p-Chloroacetophenone (-)-camphene-based** | 20 | 48.69 (2.88) | |
| **p-Methylacetophenone (-)-camphene-based** | 22 | 52.26 (2.75) | |
| **Thiophene-2-carboxyaldehyde (-)-camphene-based** | 25 | 14.38 (1.86) | |
| **Furaldehyde (-)-camphene-based** | 26 | 24.78 (1.67) | |
| **Ethylpyruvate (-)-camphene-based** | 31 | 26.95(2.03) | |
| Limonene thiosemicarbazide | 3' | 15.89 (1.35) | [21, 29] ¶ |
| Benzaldehyde R-(+)-limonene-based | 4' | 49.26 (2.34) | [21]¶ |
| *m-Nitro*benzaldehyde R-(+)- limonene based | 6' | 14.17 (1.54) | [21] ¶ |
| *p-Hydroxy*benzaldehyde R-(+)- limonene based | 9' | 24.75 (1.97) | [21] ¶ |
| Doxorubicin (reference compound) | - | 30.21 (2.12) | [30] # |

The compounds highlighted in bold are novel.

*References were evaluated for biological activities, such as antituberculosis, antibacterial, and antileishmanial activities.

¶The compounds mentioned were tested on melanoma cells of the UACC-62 lineage in ref. 12, and the reported values for **3'**, **4'**, **6'**, and **9'** were as follows: not tested, 31.6 µM, >100 µM, and >100 µM, respectively.

#The reported value for SK-MEL-37 cells was 35 µM.

and **6** elicited a similar effect at different time points (24, 48 and 72 hours of incubation, respectively; results not shown). The remaining compounds were not tested in this study of the mode of cell death induction.

## Morphological changes determined by different techniques

**Fluorescence microscopy analysis.** Fluorescence microscopy revealed morphological changes in detached cells as early as 6 hours of drug exposure for compound **8**,as depicted in Fig 4A. These changes persisted after 10 hours (Fig 4B) and 24 hours (Fig 4C). Likewise, compounds **4**, **11**, and **14** demonstrated similar effects, although not illustrated in the figure. Morphological alterations were observed with compounds **13** and **6**; however, these changes became apparent only after 48 and 72 hours of incubation, respectively (results not shown). The remaining compounds were not subjected to this testing.

**In vivo fluorescence-based caspase 3 activity analysis.** Caspase 3 activity was measured in SK-MEL-37 melanoma cells treated with 100 µM benzaldehyde (-)-camphene-based thiosemicarbazone (compound **4**) for 8 hours using a NucView$^{TM}$ kit. This kit contains a fluorogenic substrate that specifically detects activated caspases in apoptotic cells by labeling the nuclei with a green fluorescent signal. Fig 4, shows a clear distinction between the control cells and the treated cells (compare panels H *versus* J; I *versus* K respectively) These results suggest that caspase-3 is activated in living treated cells, providing further evidence that the cellular death mechanism may be primarily due to induction of apoptosis.

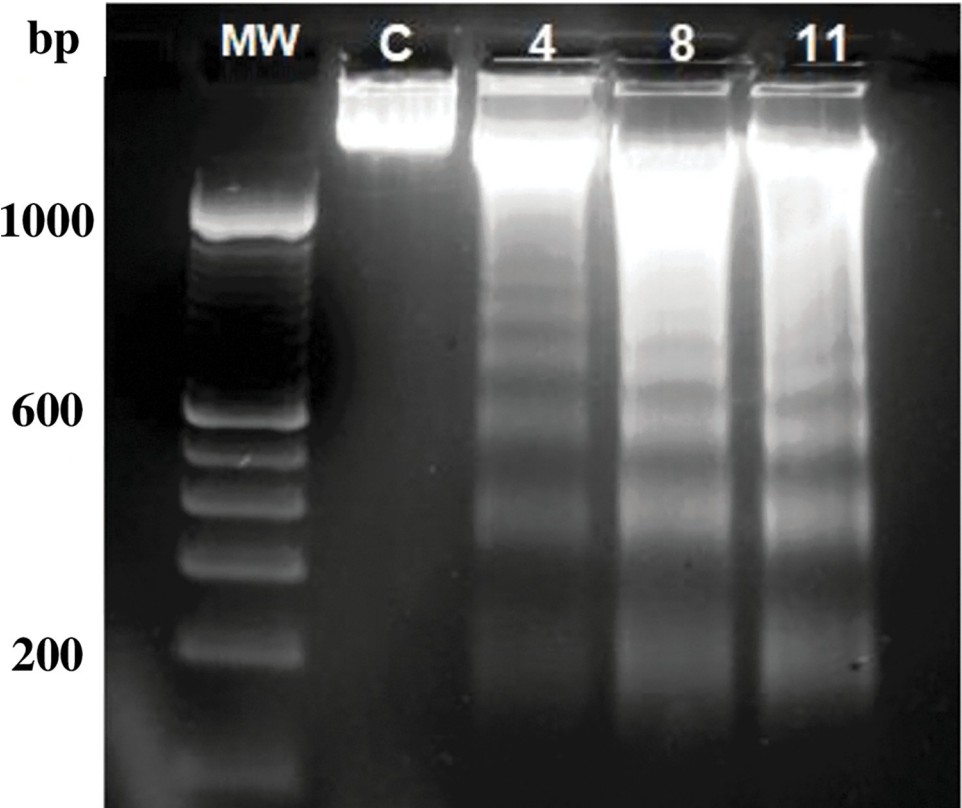

**Fig 3. Analysis of DNA fragmentation.** Exponentially growing SK-MEL-37 melanoma cells were treated with 100 μM concentrations of the indicated compounds: (**4**) benzaldehyde (-)-camphene-based thiosemicarbazone; (**8**) *m-chloro*benzaldehyde (-)-camphene-based thiosemicarbazone; and (**11**) *m-nitro*benzaldehyde(-)-camphene-based thiosemicarbazone. After 24 hours of incubation, attached and floating cells were collected, and DNA was extracted and analyzed by agarose gel electrophoresis as detailed in the Materials and Methods section. Molecular weight (MW): DNA 100 bp ladder (Invitrogen).

**Scanning electron microscopy (SEM) analysis.** SEM analysis revealed changes on the surface of cells following treatment with 100 μM benzaldehyde (-)-camphene-based thiosemicarbazone (compound **4**). Untreated cells showed numerous cytoplasmic projections (Fig 4, panels D and E), whereas cells treated with compound **4** for 8 hours exhibited a reduction in the number of these projections and the formation of vacuoles (Fig 4, panels F and G), consistent with apoptotic changes.

## Colorimetric assay for measuring the activity of caspases 2, 3, 6, 8, and 9

To gain a better understanding of the mechanism of action of the compounds under investigation, we performed a colorimetric assay to measure the activity of caspases 2, 3, 6, 8, and 9. Notably, this assay was conducted exclusively with compound **4**, which was chosen as a representative based on the $IC_{50}$ value obtained.

Fig 5 displays the results of the colorimetric assay measuring the activity of caspases 2, 3, 6, 8, and 9. Our findings indicated that caspases 6 and 8 exhibited activity within the first 2 hours of treatment with compound **4**. Only caspases 6 and 8 were significantly activated in relation to the level in the control, but the activity of caspase 8 rapidly decreased to low levels after 4 h, remaining low at 6 and 24 h of incubation. The activity of caspase 6 remained elevated, reaching its peak at 6 hours and declining at 24 hours of incubation. These results suggested that

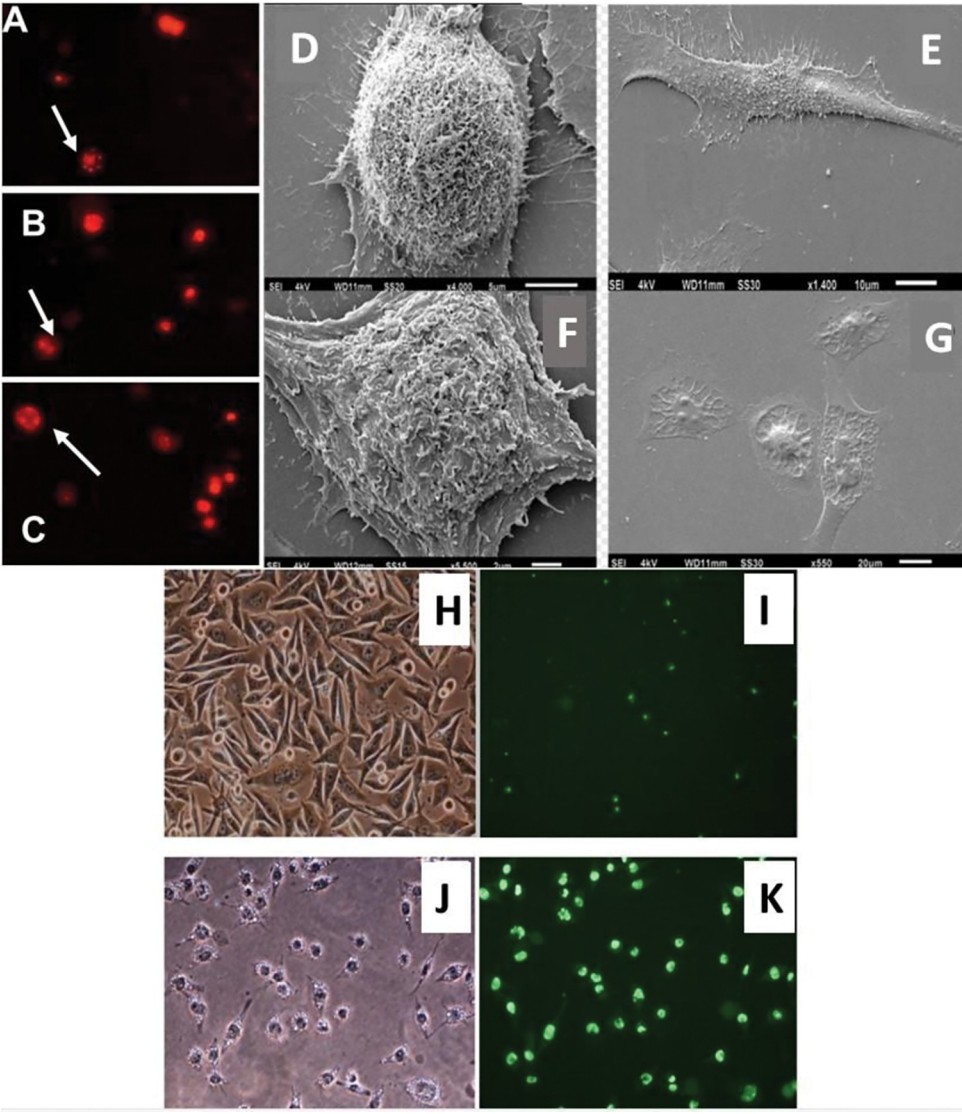

**Fig 4. Morphological assessment of apoptotic changes observed through different techniques.** Panels A to C show fluorescence microscopy images of SK-MEL-37 cells exposed to *m-chloro*benzaldehyde (-)-camphene-based thiosemicarbazone (compound **8**) at a concentration of 50 μM. Propidium iodide was added to the culture medium 4 hours after treatment and remained in the medium throughout the incubation period. After 6 hours of treatment, a few cells with fragmented nuclei were observed in panel A (indicated by arrows), and their numbers increased after 10 (panel B) and 24 hours (panel C) of treatment. The addition of propidium iodide did not affect the untreated cells. All observed cells were floating in the culture medium. Panels D to G show scanning electron microscopy images of cells cultured for 8 hours and treated with the solvent only (panels D and E) or with 100 μM benzaldehyde (-)-camphene-based thiosemicarbazone (panels F and G), which resulted in a reduction in the number of membrane projections and the appearance of vacuoles in treated cells. Panels J and K show SK-MEL-37 cells treated with benzaldehyde (-)-camphene-based thiosemicarbazone (compound **4**) at a concentration of 100 μM for 6 hours, and panels H and I display control cells. All cells were stained with NucView™ for caspase-3 activity detection using fluorescence microscopy. The images in panels H and J were captured without a fluorescent filter to enhance the visibility of morphological distinctions between control cells and treated cells. Overall, this figure demonstrates the morphological changes induced by compounds **4** and **8** in SK-MEL-37 cells as visualized through fluorescence and scanning electron microscopy.

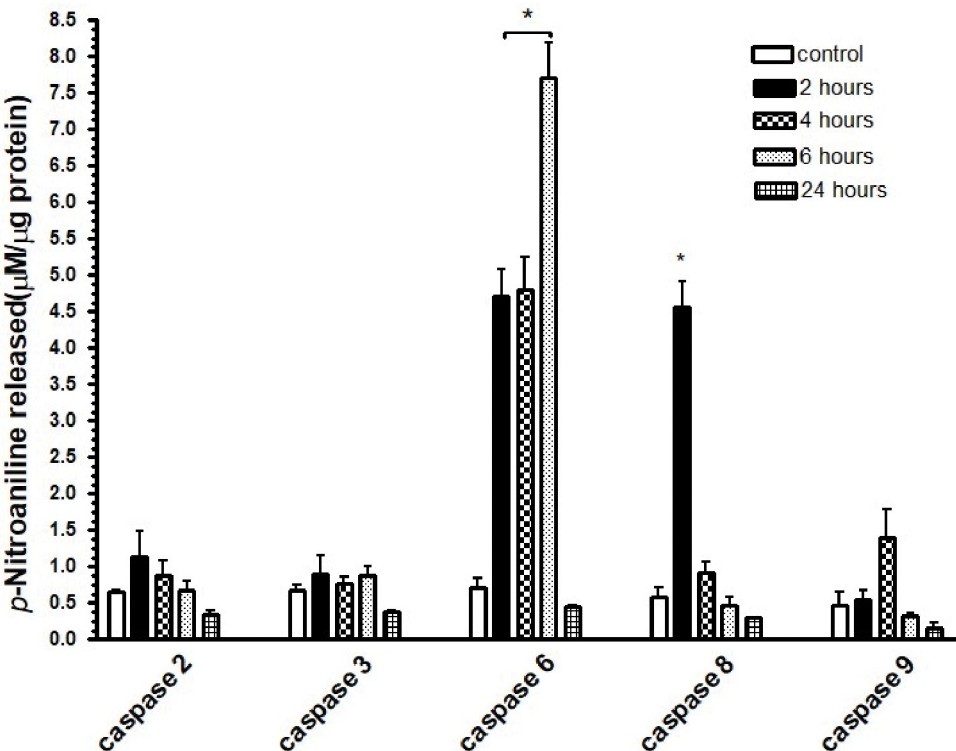

**Fig 5. Caspase 2, 3, 6, 8, and 9 activity in SK-MEL-37 melanoma cells.** Exponentially growing cells were treated with 100 μM benzaldehyde (-)-camphene-based thiosemicarbazone for 2, 4, 6, and 24 hours, as specified in the figure. The activity of caspases 2, 3, 6, 8, and 9 was determined by incubation with 200 μM peptide substrates at 37°C under reaction conditions that were standardized as detailed in the Materials and methods section. The kinetics graphs obtained can be found in S2 File. The values presented in the figure correspond to the activity obtained at the optimal incubation time, which was 15 or 30 hours. This figure displays columns representing the mean and standard deviation of duplicate wells from two independent experiments. Statistical significance (*, p < 0.05) is indicated in the figure. The values were compared with those for the control using ANOVA with a Dunnett *post hoc* test.

compound **4** could selectively activate mainly caspases 6 and 8 and that their activation is time-dependent during the period of incubation.

## Annexin V-FITC/PI staining assessment by flow cytometry

In Fig 6, treatment with benzaldehyde (-)-camphene-based thiosemicarbazone (compound 4) for just 8 hours resulted in the detection of recent and late apoptotic features in 3.7% and 62% of cells, respectively. Similarly, for benzaldehyde R-(+)-limonene-based thiosemicarbazone (compound **4'**), the percentages of cells exhibiting early and late apoptotic events were 16% and 23%, respectively. These findings suggest that both compounds have the potential to induce apoptosis in cells, although compound **4** appears to be more effective in inducing late apoptotic events than compound **4'**. The high proportion of cells in late apoptosis observed for compound **4** is consistent with the DNA fragmentation profile observed by agarose gel electrophoresis, further supporting its potency in inducing apoptosis.

## Discussion

In this study, we synthesized and structurally characterized 19 thiosemicarbazone compounds based on (-)-camphene using $^1$H and $^{13}$C-NMR techniques. The structure of these compounds is illustrated in Fig 1, and structural characterization is provided in S1 File. This is the first

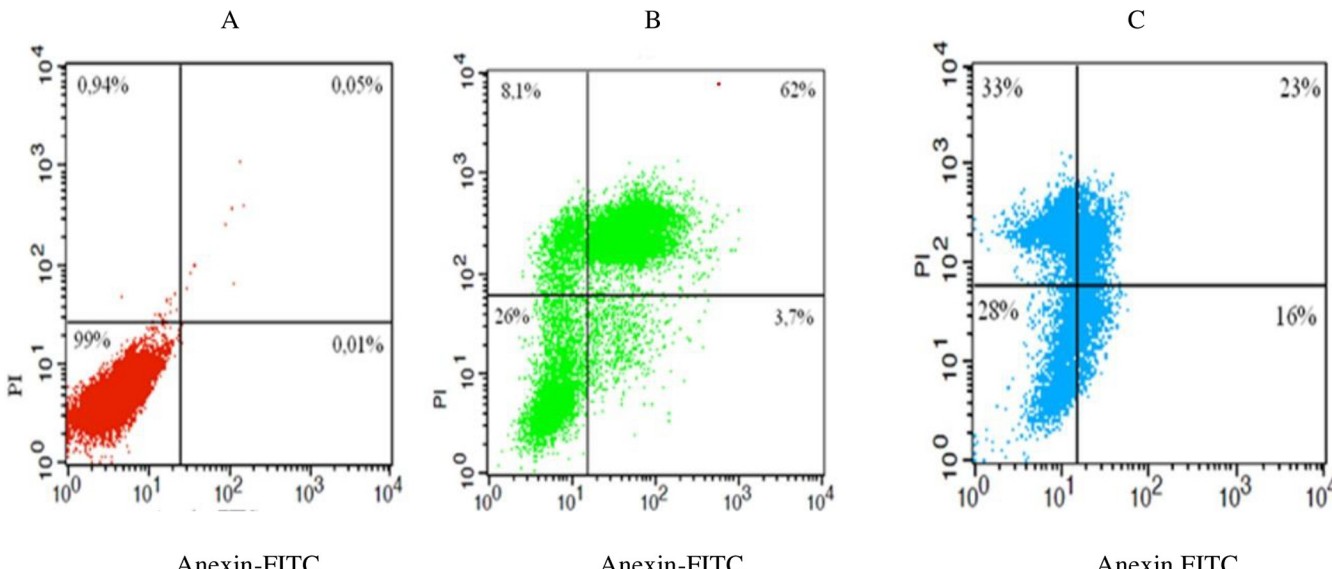

**Fig 6. Early and late apoptotic events in SK-MEL-37 melanoma cells.** Exponentially growing cells were treated with (A) solvent (DMSO), (B) 100 μM benzaldehyde (-)-camphene-based thiosemicarbazone (compound **4**) or (C) 100 μM benzaldehyde R-(+)-limonene-based thiosemicarbazone (compound **4'**). After 8 hours of treatment, the cells were labeled with Annexin and propidium iodide (PI) for flow cytometric analysis of apoptosis. The results showed that treatment with both substances induced early and late apoptotic events.

time that the biological activity of these compounds has been evaluated. The antiproliferative activity of these thiosemicarbazones was investigated in SK-MEL-37 melanoma cells. Among these newly synthesized compounds, nine compounds, *m-chloro*benzaldehyde (**8**), *m-nitro*-benzaldehyde (**11**), *m-methoxy*benzaldehyde (**15**), acetophenone (**18**), *p-chloro*acetophenone (**20**), *p-methyl*acetophenone (**22**), thiophene-2-carboxyaldehyde (**25**), furaldehyde (**26**) and ethylpyruvate (**31**) (-)-camphene-based thiosemicarbazone, exhibited antiproliferative activity, with cell viability below 20% and $IC_{50}$ values ranging from 13–55 μM. They have shown promising antiproliferative activity against human melanoma cells and warrant further evaluation in future studies.

Additionally, thirteen (-)-camphene and six R-(+)-limonene-based thiosemicarbazones (see structure in Fig 1) were previously assessed for various biological activities, including antibacterial, antifungal and antileishmanial properties. Here, they were evaluated for their antiproliferative activity against SK-MEL-37 cells, and ten compounds produced cell viability lower than 20%, with $IC_{50}$ values ranging from 11 to 49 μM.

Thus, (-)-camphene-based thiosemicarbazone containing *p-hydroxy*benzaldehyde (**14**) and *o-hydroxy*benzaldehyde (**17**) exhibited $IC_{50}$ values of 15.39 and 11.56 μM against SK-MEL-37 cells, respectively. Notably, compound **17** demonstrated the lowest $IC_{50}$ value among all the compounds examined in our study. In a previously reported study, these same two compounds were found to have the best minimal inhibitory concentration (MIC) values against *M. tuberculosis* and were also the most cytotoxic toward Vero cells, with $IC_{50}$ values of 18.5 and 6.7 mg/mL, respectively [20], equivalent to 55 μM and 20 μM, based on their molecular weight. Additionally, compound **14** was found to be active against *Staphylococcus aureus* and *Enterococcus spp.* in another study [19]. Interestingly, *p-nitro*benzaldehyde (-)-camphene-based (**12**) exhibited poor antiproliferative activity against SK-MEL-37 cells (Fig 2, left y-axis) and no anti-tuberculosis activity [20]. Importantly, the presence of antimicrobial/antifungal activities in a compound does not guarantee anticancer potential, and conversely, the absence of antimicrobial/antifungal activities does not imply the absence of anticancer activity.

When examining the antiproliferative activity against human melanoma cells of another cell line, we observed that the compound benzaldehyde R-(+)-limonene-based thiosemicarbazone (**4'**) exhibited antiproliferative activity against SK-MEL-37 cells (in this study) and UACC-62 cells in a previously published study [21], with IC$_{50}$ values of 49.26 μM and 31.6 μμM, respectively. However, the compounds *m-nitro*benzaldehyde limonene-based (**6'**) and *p-hydroxy*benzaldehyde limonene-based thiosemicarbazone (**9'**) were not active against UACC-62 melanoma cells in a previous study [21] but showed activity against SK-MEL-37 cells in our study, with IC$_{50}$ values of 14.17 and 24.75 μM, respectively.

Next, we prioritized compound **4** in our attempt to explore the mechanism of action of this class of compounds, as it exhibited one of the lowest IC$_{50}$ values among the 38 compounds evaluated and was also easily synthetized.

Using the DNA ladder assay, we observed DNA fragments that are highly indicative of apoptotic cell death. Although this technique is simple and does not require specialized equipment, it is worth noting that internucleosomal DNA fragmentation typically occurs in the latter stages of apoptotic processes [33]. Additionally, the absence of the "smear" pattern typically observed in necrotic processes suggests that necrosis is less likely to be a prominent event. These findings were further corroborated by flow cytometry, which detected a proportion of such events. Further analysis using fluorescence and scanning electron microscopy revealed numerous morphological alterations, including nuclear fragmentation and cell shrinkage. Activation of caspase 3 in living cells was also easily observed by fluorescence, further confirming the occurrence of apoptotic processes.

To identify the possible signaling pathways involved, we conducted an assay to detect the activity of caspases 2, 3, 6, 8, and 9 in cell lysates, and we found that the tested compound **4** induced the activity of caspases 6 and 8 as early as 2 hours after drug treatment. Notably, the induction of caspase 6 activity persisted for 4 hours and reached a peak after 6 hours, a trend not observed with caspase 8. The early and abrupt activation of caspase 8 suggested that compound **4** may initiate the death receptor (DR) pathway, leading to caspase 3 activation and subsequent cell death [34]. The activation of caspase 6 suggests that compound **4** may also elicit intracellular signaling, leading to the activation of the intrinsic pathway. However, importantly, despite caspase 6's well-known role as an executioner caspase in the intrinsic pathway, it may also possess an initiation function, potentially priming the cell for the receptor-mediated pathway [35]. Based on these findings, we hypothesized that the tested compound **4** may act through an extrinsic apoptotic pathway, although the involvement of the intrinsic pathway cannot be completely ruled out. Further research is needed to fully elucidate the mechanism of action not only of compound **4** but also of the preselected compounds with low IC$_{50}$ values, which was not possible due to budgetary constraints.

These findings indicated the promising potential of the investigated compounds as therapeutic agents for cancer treatment, as they exhibited IC$_{50}$ values below those found for the reference compound, doxorubicin. Further investigation into their precise mechanisms is crucial to fully comprehend their therapeutic properties. The discovery of compounds that induce apoptotic cell death holds significant value, as this mode of cell death is frequently associated with a reduced likelihood of drug resistance and improved clinical outcomes [36]. Moreover, to our knowledge, the activation of caspase 8 has not been previously observed, as the intrinsic pathway involving the formation of reactive oxygen species predominates in the mode of action of various other thiosemicarbazone derivatives, according to previous reports [37].

## Conclusions

Thiosemicarbazone derivatives, synthesized from (-)-camphene and R(+)-limonene, demonstrated significant antiproliferative activity against human metastatic melanoma cells,

emphasizing their potential for further exploration in the context of aggressive melanoma therapy. Benzaldehyde (-)-camphene-based thiosemicarbazone, devoid of substituents on the benzene ring, induced apoptotic events, as confirmed by DNA fragmentation, flow cytometry, and scanning electron microscopy assays. These findings were accompanied by observation of the activation of caspase 6 and 8 enzymatic activities, suggesting the involvement of the extrinsic pathway. Subsequent studies should encompass a variety of cancer cell lines, *in vivo* models, and docking studies to elucidate the mode of action of these compounds.

## Supporting information

**S1 File. Structural characterization of new thiosemicarbazone derivatives.** Structural characterization by $^1$H and $^{13}$C NMR of the new (-)-camphene-based compounds.
(PDF)

**S2 File. Standardization for the colorimetric enzymatic assay of caspases 2, 3, 6, 8 and 9.** Optimization of kinetic conditions for the enzymatic reaction of caspases.
(PDF)

**S1 Raw image. Raw electrophoresis gel image.**
(PDF)

## Acknowledgments

We are grateful to Elisangela Ribeiro, Immunophenotyping Laboratory, Cancer Combat Association in Goiás, Brazil, for help with and expertise in flow cytometry. We thank Dr. Fabio Vandresen, State University of Maringá, for suggestions regarding manuscript organization.

## Author Contributions

**Conceptualization:** Cecília Maria Alves de Oliveira, Lucília Kato, Cleuza Conceição da Silva, Lidia Guillo.

**Funding acquisition:** Lidia Guillo.

**Investigation:** Paula Roberta Otaviano Soares, Débora Cristina Souza Passos, Francielly Moreira da Silva, Ana Paula B. da Silva-Giardini, Narcimário Pereira Coelho.

**Methodology:** Cecília Maria Alves de Oliveira, Lucília Kato, Lidia Guillo.

**Resources:** Cecília Maria Alves de Oliveira, Lucília Kato, Lidia Guillo.

**Visualization:** Lidia Guillo.

**Writing – original draft:** Cecília Maria Alves de Oliveira, Lidia Guillo.

**Writing – review & editing:** Lidia Guillo.

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
