## [Decision Letter · Decision Letter 0]

30 Aug 2023

PONE-D-23-14168In vitro antiproliferative and apoptotic effects of thiosemicarbazones based on (-)-camphene and R-(+)-limonene in human melanoma cellsPLOS ONE

Dear Dr. Guillo,

Thank you for submitting your manuscript to PLOS ONE. After careful consideration, we feel that it has merit but does not fully meet PLOS ONE’s publication criteria as it currently stands. Therefore, we invite you to submit a revised version of the manuscript that addresses the points raised during the review process.

We look forward to receiving your revised manuscript.

Kind regards,

Wagdy Mohamed Eldehna, Ph.d

Academic Editor

PLOS ONE

Journal Requirements:

"Conselho Nacional de Desenvolvimento Científico e Tecnológico"

One or more authors are affiliated with the funder, but authors state that the funder had no role

Thank you for stating the following financial disclosure:

"Authors  PROS and DCSP were supported by a scholarship from the National Council for Scientific and Technological Development (CNPq).

The funders had no role in study design, data collection and analysis, decision to publish, or preparation of the manuscript "

We note that one or more of the authors is affiliated with the funding organization, indicating the funder may have had some role in the design, data collection, analysis or preparation of your manuscript for publication; in other words, the funder played an indirect role through the participation of the co-authors. If the funding organization did not play a role in the study design, data collection and analysis, decision to publish, or preparation of the manuscript and only provided financial support in the form of authors' salaries and/or research materials, please do the following:

           1. Review your statements relating to the author contributions, and ensure you have specifically and accurately indicated the role(s) that these authors had in your study. These amendments should be made in the online form.

          2. Confirm in your cover letter that you agree with the following statement, and we will change the online submission form on your behalf:

“The funder provided support in the form of salaries for authors [insert relevant initials], but did not have any additional role in the study design, data collection and analysis, decision to publish, or preparation of the manuscript. The specific roles of these authors are articulated in the ‘author contributions’ section.

Reviewers' comments:

Reviewer's Responses to Questions

**Comments to the Author**

1. Is the manuscript technically sound, and do the data support the conclusions?

Reviewer #1: No

Reviewer #2: Yes

2. Has the statistical analysis been performed appropriately and rigorously? 

Reviewer #1: Yes

Reviewer #2: Yes

3. Have the authors made all data underlying the findings in their manuscript fully available?

Reviewer #1: No

Reviewer #2: Yes

4. Is the manuscript presented in an intelligible fashion and written in standard English?

Reviewer #1: No

Reviewer #2: No

5. Review Comments to the Author

Reviewer #1: Dear editor,

Have a nice day. The authors synthesized some thiosemicarbazone derivatives as anticancer agents. The synthesized compounds were evaluated in vitro for their anti-proliferative effect.

The manuscript need extensive modifications as follows.

1- The language of the manuscript should be extensively revised.

2- Some symbols and words should be italic form as heteroatoms of the chemical names and in vitro.

3- Abstract should be more concise.

4- Some words start with capital letters although being in the middle of the sentence. Please, check all manuscript.

5- The introduction section is not comprehensive. It should comprise the background, aim of the work, the problems, and the rational of the work.

6- 1H NMR and 13 C NMR charts should be provided in supplementary data.

7- Fig. 1, 2, 3 have poor resolution.

8- The biological target (receptor or enzyme) should be determined. Then docking studies should be carried out against the target receptor.

Reviewer #2: The work exerted is so appreciated. However, some points need to be addressed. So, a major revision may be required to improve the manuscript.

1- The abstract has to be improved. The abstract should illustrate your work in an attractive way and briefly removing all redundant data.

2- Some typos and grammatical errors need to be corrected.

3- In the introduction part, some short paragraphs need to be united into one paragraph.

4- The overall introduction is too short. Many points have to be discussed in the introduction part such as impact of cancer and its global statistics, attempts for melanoma treatment, literature of compounds bearing thiosemicarbazone for cancer treatment, and so on….

5- “Please note that in the manuscript, all compounds will be highlighted in bold for readability purposes.” Please remove this sentence!

6- “characterized by one of the coauthors”. Please remove this sentence!

7- At lines 377-379 and 459-471, please remove highlights.

8- The resolution of most figures have to be enhanced.

9- The experimental chemistry part of the synthesized compounds along with chemical synthesis procedures should be transferred from supplementary to the main manuscript.

10- Conclusion part have to be improved.

11- Some references have to be updated.

12- Where are the charts of IR, 1HNMR, 13CNMR and mass spectrometry?? They have to be shown in the supplementary material.

13- The work presented is so valued. However, the overall writing pattern of the manuscript has to be improved.

6. PLOS authors have the option to publish the peer review history of their article (what does this mean?). If published, this will include your full peer review and any attached files.

Reviewer #1: **Yes: **Ibrahim Eissa

Reviewer #2: No

---

## [Author Response · Author response to Decision Letter 0]

23 Oct 2023

Reviewers #1 and #2: We are pleased to inform you that all the suggested revisions and modifications have been meticulously addressed, and you can find our detailed responses in the file labeled 'Response to Reviewers'. Thank you both for dedicating your time and effort in reviewing our work. Your contributions have been invaluable in improving the quality and clarity of our manuscript.

---

## [Decision Letter · Decision Letter 1]

14 Nov 2023

In vitro antiproliferative and apoptotic effects of thiosemicarbazones based on (-)-camphene and R-(+)-limonene in human melanoma cells

PONE-D-23-14168R1

Dear Dr. Guillo,

We’re pleased to inform you that your manuscript has been judged scientifically suitable for publication and will be formally accepted for publication once it meets all outstanding technical requirements.

Kind regards,

Wagdy Mohamed Eldehna, Ph.d

Academic Editor

PLOS ONE

Additional Editor Comments (optional):

Reviewers' comments:

Reviewer's Responses to Questions

**Comments to the Author**

1. If the authors have adequately addressed your comments raised in a previous round of review and you feel that this manuscript is now acceptable for publication, you may indicate that here to bypass the “Comments to the Author” section, enter your conflict of interest statement in the “Confidential to Editor” section, and submit your "Accept" recommendation.

Reviewer #1: All comments have been addressed

Reviewer #2: All comments have been addressed

2. Is the manuscript technically sound, and do the data support the conclusions?

Reviewer #1: Yes

Reviewer #2: Partly

3. Has the statistical analysis been performed appropriately and rigorously? 

Reviewer #1: N/A

Reviewer #2: N/A

4. Have the authors made all data underlying the findings in their manuscript fully available?

Reviewer #1: Yes

Reviewer #2: Yes

5. Is the manuscript presented in an intelligible fashion and written in standard English?

Reviewer #1: Yes

Reviewer #2: Yes

6. Review Comments to the Author

Reviewer #1: Dear editor,

Have a nice day. The authors carried out all the requested modifications. Accordingly, I recommend the publication of the manuscript in the current form.

Best regards

Reviewer #2: The paper entitled by "In vitro antiproliferative and apoptotic effects of thiosemicarbazones 1 based on (-)-

2 camphene and R-(+)-limonene in human melanoma cells" can be accepted

7. PLOS authors have the option to publish the peer review history of their article (what does this mean?). If published, this will include your full peer review and any attached files.

Reviewer #1: No

Reviewer #2: No

---

## [Editor Report · Acceptance letter]

20 Nov 2023

PONE-D-23-14168R1 

*In vitro* antiproliferative and apoptotic effects of thiosemicarbazones based on (-)-camphene and R-(+)-limonene in human melanoma cells 

Dear Dr. Guillo:

I'm pleased to inform you that your manuscript has been deemed suitable for publication in PLOS ONE. Congratulations! Your manuscript is now with our production department. 

Kind regards, 

on behalf of

Dr. Wagdy Mohamed Eldehna 

Academic Editor

PLOS ONE